# Food consumption diversity and nutritional status among children aged 6–23 months in Indonesia: The analysis of the results of the 2018 Basic Health Research

Omas Bulan Samosir[ID]*[☯], Dinda Srikandi Radjiman[☯], Flora Aninditya[ID][☯]

Lembaga Demografi, Faculty of Economics and Business, Universitas Indonesia, Depok, West Java, Indonesia

☯ These authors contributed equally to this work.
* omasbr@yahoo.co.uk

**Data Availability Statement:** Data cannot be shared publicly because the data belong to the National Institute of Health Research and

## Abstract

### Background

Stunting among children under five years of age is among the highest in Indonesia. The objective of this study was to investigate the association between food consumption diversity and nutritional status among children aged 6–23 months in Indonesia.

### Methods

The data used came from the results of the 2018 Basic Health Research. The main independent variable was the diversity of food consumption. Control variables included breastfeeding practices and demographic and socioeconomic factors. Using ordinal logistic regression, the role of food consumption diversity in influencing nutritional status was examined after controlling for breastfeeding practices and demographic and socioeconomic factors.

### Results

The results of the study showed that the diversity of food consumption (AOR = 1.15; 95%CI: 1.07–1.24) significantly and statistically influenced nutritional status of children age 6–23 months in Indonesia even after controlling for the effects of breastfeeding practices and demographic and socioeconomic factors. Higher odds of having normal nutritional status compared to being stunted or severely stunted was associated with consuming food according to the dietary diversity. Higher odds of having normal nutritional status compared to being stunted or severely stunted was also significantly and statistically associated with being ever breastfed (AOR = 1.33; 95%CI: 1.22–1.46), aged 6–11 months (AOR = 3.07; 95%CI: 2.79–3.38), female (AOR = 1.35; 95%CI: 1.25–1.46), children of non-working mothers (AOR = 1.12; 95%CI: 1.04–1.21), children of higher educated mothers (AOR = 1.50; 95%CI: 1.30–1.72), children from higher wealth quintile households (AOR = 1.65; 95%CI:

Development (NIHRD) of the Ministry of Health of the Republic of Indonesia. Data request may be made to NIHRD through email with the address: tu. bkpk@litbang.kemkes.go.id

**Funding:** This research was supported by the Government of Indonesia, which provided for funding for this research project through the Ministry of Research and Technology/National Research and Innovation Board Fiscal Year 2021 (No.: NKB-006/UN2.RST/HKP.05.00/2021). The funder had no role in study design, data collection and analysis, decision to publish, or preparation of the manuscript.

**Competing interests:** The authors have declared that no competing interests exist.

1.44–1.85), children from smaller size family (AOR = 1.07; 95%CI: 1.05–1.10), and urban children (AOR = 1.16; 95%CI:1.08–1.25).

## Conclusion

A profound percentage of children aged 6–23 months in Indonesia experienced stunting and severely stunting. Children who did not receive minimum dietary diversity were more likely to suffer from stunting. The findings from this study suggest that to ensure the achievement of national goal of preventing stunting and sustainable development goal of ending all forms of malnutrition in Indonesia, the strategy should promote the fulfillment of minimum food consumption diversity.

## Introduction

The prevalence of stunting among children under 5 years of age is a key health problem in many developing countries including Indonesia. Although the prevalence of stunting among children aged under five years has decreased from 33.1% in 2000 to 22.0% in 2020, there were still almost 150 million children under 5 years of age in the world who experienced stunting [1]. The World Health Organization (WHO) reported that in 2020 more than half (53%) of these stunted children came from Asia [1]. In the Southeast Asia region, the prevalence of stunting was the fourth highest in Indonesia after Timor-Leste, Lao, and Cambodia [1].

The results of the 2018 Basic Health Research (*Riset Kesehatan Dasar*/RISKESDAS) showed that 31% of children under five years in Indonesia experienced stunting [2]. Not only affecting the health of the children, stunting also may lead to increased morbidities in later life [3]. In response, the Government of Indonesia is committed to reducing stunting and includes stunting reduction as one of the national strategic priorities in The National Medium-Term Development Plan For 2020–2024 [4] and issued a document on the National Strategy for the Acceleration of Prevention of Stunting for the 2018–2024 period [5]. Based on this national strategy and sustainable development goal of ending all forms of malnutrition [6], adequate food intake and nutrition are the specific nutrition interventions that target the causes of stunting. Therefore, in an effort to reduce the prevalence of stunting, an understanding of the relationship between feeding practices and the incidence of stunting in Indonesia is important.

Under nutrition in infants and young children, such as stunting can be affected by some factors. These include poor feeding practices together with household and family factors and breastfeeding practices [7]. Poor infant and young children feeding include poor micronutrient quality in food, low dietary diversity, and inadequate feeding practices, such as infrequent feeding, inadequate feeding during and after illness, thin food consistency, or feeding insufficient quantities [8]. Individual, community, and social factors were proven as stunting determinants in Indonesia [8]. Infant nutrition is essential for children's health and development during childhood and also during adolescence and adulthood [9]. Prior to the age of two, optimal breastfeeding and sufficient supplementary feeding habits for infants and young children are also protective factors against stunting and wasting [10, 11].

The results of the 2017 Indonesia Demographic and Health Survey (DHS) showed that there were still 4.8% of children aged 6–23 months who were not breastfed and not given formula milk or other dairy products [12]. In addition, 25.9% of children did not eat more than 3–4 types of complementary foods (breast milk, formula, cheese/yogurt, or other dairy products), 44.5% of children did not eat according to the minimum frequency according to their

age, and 57.7% of children aged 6–23 months did not receive all three variations of these infant and young child feeding (IYCF) practices. Suboptimal feeding practices and a high prevalence of underweight, wasting, and stunting were found among children aged 6–23 months [11, 13]. A study in Aceh showed that of 39% of children aged 6–23 months who received exclusive breastfeeding, only 61% received long-term breastfeeding, and 50% received timely complementary feeding. The minimum eating frequency was met by 74% of the subjects, but the minimum acceptable dietary diversity and diet were realized by only 50% and 40% respectively. The prevalence of underweight, wasting, and stunting found in this study were 26%, 23%, and 28% respectively [14]. A study in Myanmar showed the crucial role of breastfeeding in the nutritional status of children [15]. Breastfeeding reduces the likelihood of infants and children with stunting status as the lactose in breast milk is effective for absorbing calcium in the body so that it helps the baby's growth [10].

In 2015, a literature study on 48 documents related to feeding patterns, nutrition, factors, determinants, and practices was carried out [16]. A wide range of information and data were analyzed using current recommendations in feeding practices. Most children aged 6–8 months received complementary foods. Most children aged 12–15 months and 20–23 months were still breastfed although breastfeeding practices declined with increasing age. Most children aged 6–23 months received three meals per day (an indicator for assessing minimum meal frequency). However, in aggregate, the national figure was found to be lower.

The results of the 2007 Indonesia DHS showed that 12% of children who were not breastfed and 67% of children who were breastfed benefited from minimum feeding frequency or receiving the recommended/adequate amount of food each day [17]. The proportion of non-breastfed children who benefited from a minimum feeding frequency was much higher in the 2012 Indonesia DHS [18]. The results of the 2007 and 2012 Indonesia DHS showed that the adequacy of daily feeding frequency decreased with increasing age [17, 18]. The cessation of breastfeeding in infants is related to age and the provision of additional food to children [19]. Problems with child growth and development will occur if breastfeeding, stopped or continued, is not accompanied by adequate breastfeeding and complementary feeding according to the age of the child. With increasing age, the nutritional needs of children will also increase. If a child does not receive adequate complementary foods according to his age and nutritional needs, linear growth retardation may occur [19].

A study conducted in Klaten showed that the dominant factor of stunting was food diversity. It was also found that boys had a greater chance of suffering from nutritional problems, such as protein energy deficiency (PEM), than girls [20]. The better the provision of complementary feeding, the more the baby's weight would increase. Giving complementary feeding is said to be appropriate if the mother continues to breastfeed her baby and provides food with sufficient frequency; namely three times a day, and the mother pays attention to cleanliness when serving food. The relationship between the amount of food, type of food, and diet with nutritional status in children under five years, was also found in other studies [20, 21]. In Tanzania, the average amount of food consumed by children with growth retardation was low. The study concluded that low meal frequency, inadequate nutritional intake, small portion sizes, and limited variety of foods prevented children from meeting daily nutritional requirements. In addition, most of the mothers in this study rarely gave their children vegetables and protein, such as beans [8]. Inadequate amounts of foods, such as legumes, meat, sardines, and vegetables could prevent children under five years from meeting their nutritional needs [22].

The study on the association between the food consumption diversity and nutritional status of children aged 6–23 months in Indonesia is still limited. This is particularly due to the availability of the data at the national level. Therefore, the objective of this study was to investigate the association between the food consumption diversity and nutritional status of children aged

6–23 months in Indonesia controlling for the effects of breastfeeding practices and demographic and socioeconomic factors.

## Methods

### Data

The data source of this study came from the results of the Basic Health Research (*Riset Kesehatan Dasar*/RISKESDAS) in 2018 conducted by the Ministry of Health (MoH) of the Republic of Indonesia (RI). It was a national scale survey with two-stage sampling: first, by selecting clusters using "probability proportional to size" method, and second, by selecting households in each cluster with systematic sampling [2].

In this study, the unit of analysis was individual children aged 6–23 months who had information on the diversity of their food consumption. In RISKESDAS 2018, dietary diversity was measured in infants and children with a variety of characteristics: "currently breastfeeding or other liquids", "ever breastfed", "never breastfed", and those with "no information on breastfeeding status."

The samples were chosen chronologically and in accordance with the RISKESDAS 2018 questionnaire's skipping pattern. On question B10K32, "Has [NAME] ever been nursed or breastfed?" there were 15,518 children who provided answers on their breastfeeding status. 14,536 children out of 15,518 answered "Yes," 858 answered "No," and 124 answered "Don't know." Those who answered "don't know" on whether or not they had been breastfed were excluded from the study since they couldn't be compared to individual children who had a history of breastfeeding. Furthermore, 10,977 children and infants out of 14,536 were still receiving breast milk at the time of sampling (thus answering "Yes" to question B10K38, "Is [NAME] currently nursed or breastfed?"). In the meantime, 3,558 children and infants had received breast milk but were no longer consuming it. Individual infants and children who were still getting breast milk were asked if they had eaten or drunk anything other than breast milk in the previous 24 hours (Table 1). Therefore, individual infants and children who were still receiving breast milk without supplementary food or drink did not have information regarding their dietary diversity, which made them excluded from the sample. The sample selection flow is presented in Fig 1.

Based on the sample selection above, the number of samples in this study was as follows.

### Variables

The dependent variable in this analysis was the nutritional status (stunting), which was measured according to WHO Child Growth Standards [23]. It is the ratio of the child's height to age converted to $Z$-score. A child with $Z$-score is less than –3 is considered severely stunted, with $Z$-score is between –3 and –2 is considered stunted, and with $Z$-score is greater –2 is considered normal. The main independent variable in this study was the infant and young child feeding (IYCF) practices. Referring to the WHO definition, feeding practice was measured by the minimum dietary diversity and minimum meal frequency, with different standards for infants and children receiving breast milk and those who did not. However, due to the availability of data in RISKESDAS 2018, the main independent variable used for the indicator of IYCF practice was the diversity of food consumption only. Other independent variables used as control variables were breastfeeding practices and demographic and socioeconomic factors (mother's age, education, and working status, place of residence, age and gender of children, and number of household members) from the results of RISKESDAS 2018, meanwhile household monthly expenditure was from the results of 2018 National Socioeconomic Survey

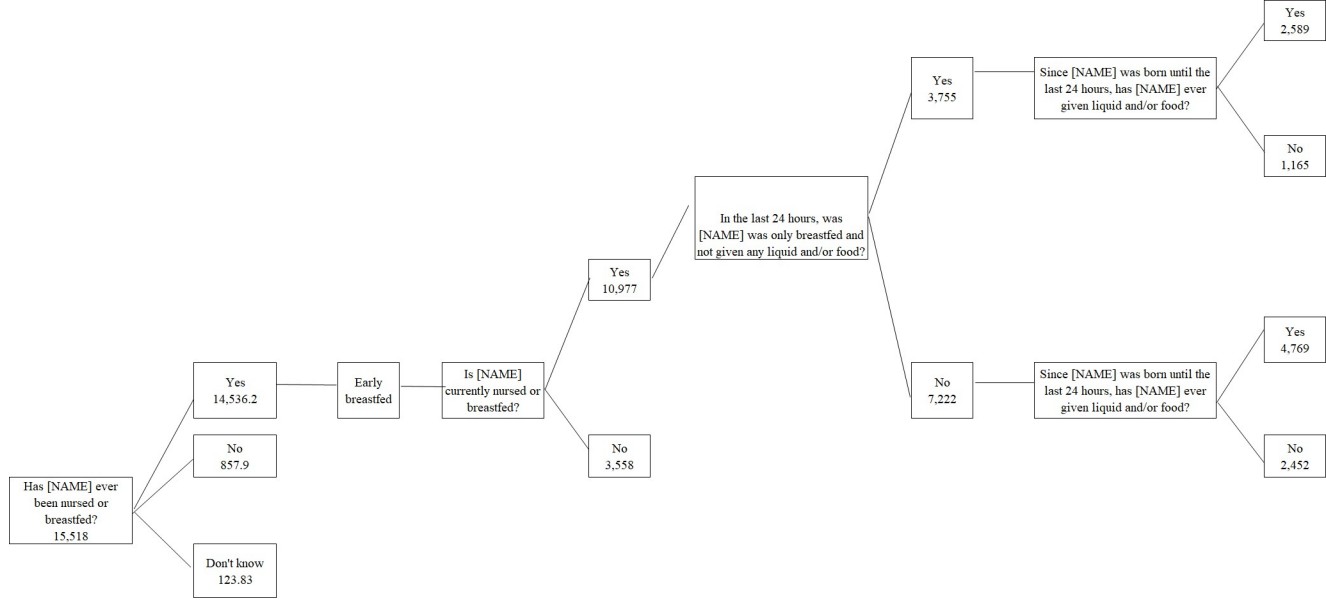

**Fig 1. Sample selection.**

(SUSENAS) data (household respondents in RISKESDAS 2018 were the same as household respondents in the SUSENAS 2018).

## Statistical analysis

The regression analysis model used was ordinal logistic regression [24]. The model was as follows.

$$\log\left[\frac{P(st \leq k)|x)}{1 - P(st \leq k|x)}\right] = \beta_0 + \beta_1 nt_{mdd} + \beta_2 earlbr + \beta_3 brstatus + \beta_4 childage$$

$$+\beta_5 childsex + \beta_6 motherage + \beta_7 motherwork + \beta_8 mothereduc$$

$$+\beta_9 hhxp + \beta_{10} hhmember + \beta_{11} residence + \varepsilon$$

$P(Y \leq k)| x)$ was the cumulative probability of children age 6–23 months having normal nutritional status with the category $k$ where $k = 2$ (normal), the $nt\_mdd$ variable was the main independent variable, namely the fulfillment of the minimum dietary diversity score, and $st$ was the nutritional status variable. The data was analyzed using STATA 15.

Control independent variables included in the analysis were the status of early initiation of breastfeeding, breastfeeding status, child's age, child's sex, mother's age, mother's working

**Table 1. Number of observations according to sample selection.**

| No. | Description | Number of observations |
|---|---|---|
| 1 | Infants and children who currently breastfed or receive foods or other liquids in the last 24 hours | 9,812 |
| 2 | Infants and children ever breastfed | 3,558 |
| 3 | Infants and children who never breastfed | 858 |
| | **Total number of observations** | **14,227** |
| | **Total number of weighted observations** | **14,216** |

status, mother's education, household's monthly expenditure quintile that was a proxy of household's wealth, and the place of residence.

## Ethics statement

The RISKESDAS in 2018 was conducted by the Ministry of Health (MoH) Republic of Indonesia. The survey implementation was reviewed and approved by the Ethics Committee of the National Institute of Health Research and Development (NIHRD) of the MoH of the RI. Written informed consent was obtained from all participants for inclusion in RISKESDAS. The participants of RISKESDAS were informed about RISKEDAS and the expected responses from them. After receiving the information about RISKESDAS, the participants confirmed their voluntary participation by signing the inform consent form. For young respondents aged less than 17 years old, the inform consent form was signed by their legal guardian. The data used in the analysis was obtained from MoH.

## Results

The sample consisted of 14,216 children aged 6–23 months. Their mean age was 14.56 months (SD 5.24 months). The mean age of mothers was 30.24 years (SD 6.38 years). The characteristics of the sample in the study was given in Table 2. It can be seen that a substantial percentage (32%) of children in the study were stunted or severely stunted, more than half did not meet minimum dietary diversity, 60% received early initiation of breastfeeding, 69% currently breastfed, 34% aged 6–11 months old, half girls, 48% of mothers aged 20–29 years old, 64% of mothers who did not work, 61% of mothers who had junior or senior high school education, 22% came from households with highest monthly expenditure quintile, 70% came from households with less than six members, and more than half lived in urban areas.

The percentage distribution of nutritional status of the children in the study by the background characteristics was displayed in Table 3. It can be seen that the percentage of severely stunted children differed by the children's background characteristics. It was higher among children who did not meet the minimum dietary diversity, did not receive early initiation of breastfeeding, were never breastfed, were aged 18–23 months old, were boys, were of teenage mothers, were of working mothers, were of lower educated mothers, were of poorer households, were from larger size households, and who lived in rural areas.

The results of ordinal logistic regression of the association between feeding practices and control variables and nutritional status in terms of the odds ratio of having normal nutritional status compared to stunted and severely stunted were presented in Table 4. It can be seen that the fulfillment of the minimum dietary diversity was statistically a significant factor in determining the nutritional status of infants and children. For infants and children, the chance of experiencing normal nutritional status compared to experiencing severely stunted and stunted status was found to be 1.15 (95%CI: 1.07–1.24) times higher for those who met the minimum dietary diversity.

The results of the study show that early initiation of breastfeeding statistically had no significant effects on the nutritional status of children. Meanwhile, breastfeeding status statistically had a substantial impact on nutritional status. Infants and children who were ever breastfed had a greater likelihood (AOR = 1.33; 95%CI: 1.22–1.46) of having normal nutritional status than being stunted and severely stunted compared to infants and children who were never breastfed.

The nutritional condition of children was found to be influenced by demographic and socioeconomic factors. Significantly and statistically, infants and children aged 6–11 months had a 3.07 (95%CI: 2.79–3.38) times higher likelihood of having a normal nutritional status

**Table 2. Percentage distribution of infants and children aged 6–23 months by background characteristics, RIS-KESDAS 2018.**

| Background characteristics | Number of Observation | % |
|---|---:|---:|
| **Nutritional Status** | | |
| Severely stunted | 1,888 | 13.3 |
| Stunted | 2,658 | 18.7 |
| Normal | 9,669 | 68.0 |
| **Meet Minimum Dietary Diversity** | | |
| No | 7,690 | 54.0 |
| Yes | 6,526 | 46.0 |
| **Early Initiation of Breastfeeding** | | |
| No | 5,741 | 40.4 |
| Yes | 8,475 | 59.6 |
| **Breastfeeding Status** | | |
| Currently breastfed | 9,803 | 69.0 |
| Ever breastfed | 3,555 | 25.0 |
| Never breastfed | 857 | 6.0 |
| **Child's Age** | | |
| 6–11 months | 4,851 | 34.1 |
| 12–17 months | 4,505 | 31.7 |
| 18–23 months | 4,861 | 34.2 |
| **Child's Sex** | | |
| Male | 7,085 | 49.8 |
| Female | 7,131 | 50.2 |
| **Mother's Age** | | |
| <20 years | 944 | 6.6 |
| 20–29 years | 6,846 | 48.2 |
| 30+ years | 6,425 | 45.2 |
| **Mother's Working Status** | | |
| Not Working | 9,069 | 63.8 |
| Working | 5,147 | 36.2 |
| **Mother's Education** | | |
| Never Attended School/Primary School | 3,805 | 26.8 |
| Junior/Senior High School | 8,624 | 60.7 |
| Higher Education | 1,786 | 12.6 |
| **Household Monthly Expenditure Quintile** | | |
| Lowest | 2,766 | 19.5 |
| Second | 2,769 | 19.5 |
| Middle | 2,754 | 19.4 |
| Fourth | 2,776 | 20.4 |
| Highest | 3,152 | 22.2 |
| **Number of Household Members** | | |
| < 6 people | 10,005 | 70.4 |
| 6–10 people | 4,106 | 28.9 |
| > 10 people | 104 | 0.7 |
| **Place of Residence** | | |
| Urban | 7,530 | 53.0 |
| Rural | 6,686 | 47.0 |
| **Total** | 14,216 | 100.0 |

than severely stunted and stunted status than children aged 18–23 months. Furthermore, compared to boys, significantly and statistically, female infants and girls had a 1.35 (95%CI: 1.25–

**Table 3. Percentage distribution of nutritional status of infants and children aged 6–23 months by background characteristics, RISKESDAS 2018.**

| Background characteristics | Nutritional Status | | | | | | | |
|---|---|---|---|---|---|---|---|---|
| | Severely Stunted | | Stunted | | Normal | | Total | |
| | Obs. | % | Obs. | % | Obs. | % | Obs. | % |
| **Meet Minimum Dietary Diversity** | | | | | | | | |
| No | 1,077 | 14.0 | 1,389 | 18.1 | 5,224 | 67.9 | 7,690 | 100.0 |
| Yes | 811 | 12.4 | 1,269 | 19.4 | 4,446 | 68.1 | 6,526 | 100.0 |
| **Early Initiation of Breastfeeding** | | | | | | | | |
| No | 777 | 13.5 | 1,109 | 19.3 | 3,856 | 67.2 | 5,741 | 100.0 |
| Yes | 1,112 | 13.1 | 1,549 | 18.3 | 5,813 | 68.6 | 8,475 | 100.0 |
| **Breastfeeding Status** | | | | | | | | |
| Currently breastfed | 1,334 | 13.6 | 1,869 | 19.1 | 6,601 | 67,2 | 9,803 | 100.0 |
| Ever breastfed | 435 | 12.3 | 626 | 18.3 | 2,494 | 68.6 | 3,555 | 100.0 |
| Never breastfed | 120 | 14.0 | 163 | 18.7 | 574 | 68.0 | 857 | 100.0 |
| **Child's Age** | | | | | | | | |
| 6–11 months | 429 | 8.8 | 573 | 11.8 | 3,849 | 79.3 | 4,851 | 100.0 |
| 12–17 months | 578 | 12.8 | 937 | 20.8 | 2,990 | 66.4 | 4,505 | 100.0 |
| 18–23 months | 882 | 18.1 | 1,148 | 23.6 | 2,831 | 58.2 | 4,861 | 100.0 |
| **Child's Sex** | | | | | | | | |
| Male | 1,105 | 15.6 | 1,342 | 18.9 | 4,638 | 65.5 | 7,085 | 100.0 |
| Female | 784 | 11.0 | 1,316 | 18.5 | 5,031 | 70.6 | 7,131 | 100.0 |
| **Mother's Age** | | | | | | | | |
| <20 years | 147 | 15.6 | 182 | 19.2 | 615 | 65.2 | 944 | 100.0 |
| 20–29 years | 901 | 13.2 | 1,360 | 17.7 | 4,731 | 69.1 | 6,846 | 100.0 |
| >30 years | 840 | 13.1 | 960 | 19.7 | 4,322 | 67.3 | 6,425 | 100.0 |
| **Mother's Working Status** | | | | | | | | |
| Not Working | 1,174 | 12.9 | 1,678 | 18.5 | 6,218 | 68.6 | 9,069 | 100.0 |
| Working | 715 | 13.9 | 981 | 19.1 | 3,452 | 67.1 | 5,147 | 100.0 |
| **Mother's Education** | | | | | | | | |
| Never Attended School / Primary School | 651 | 17.1 | 817 | 21.5 | 2,337 | 61.4 | 3,805 | 100.0 |
| Junior/Senior High School | 1,075 | 12.5 | 1,571 | 18.2 | 5,978 | 69.3 | 8,624 | 100.0 |
| Higher Education | 162 | 9.1 | 270 | 15.1 | 1,353 | 75.8 | 1,786 | 100.0 |
| **Household Monthly Expenditure Quintile** | | | | | | | | |
| Lowest | 433 | 15.7 | 625 | 22.6 | 1,709 | 61.8 | 2,766 | 100.0 |
| Second | 449 | 16.2 | 542 | 19.6 | 1,778 | 64.2 | 2,769 | 100.0 |
| Middle | 395 | 14.3 | 551 | 20.0 | 1,808 | 65.7 | 2,754 | 100.0 |
| Fourth | 332 | 12.0 | 456 | 16.4 | 1,987 | 71.6 | 2,776 | 100.0 |
| Highest | 279 | 8.9 | 484 | 15.4 | 2,388 | 75.8 | 3,152 | 100.0 |
| **Number of Household Members** | | | | | | | | |
| <6 people | 1,291 | 12.9 | 1,845 | 18.4 | 6,870 | 68.7 | 10,005 | 100.0 |
| 6–10 people | 578 | 14.1 | 802 | 19.5 | 2,726 | 66.4 | 4,106 | 100.0 |
| >10 people | 21 | 20.2 | 12 | 11.5 | 72 | 69.2 | 104 | 100.0 |
| **Place of Residence** | | | | | | | | |
| Urban | 846 | 11.2 | 1,344 | 17.9 | 5,340 | 70.9 | 7,530 | 100.0 |
| Rural | 1,043 | 15.6 | 1,314 | 19.7 | 4,329 | 64.8 | 6,686 | 100.0 |
| **Total** | 1,888 | 13.3 | 2,658 | 18.7 | 9,669 | 68.0 | 14,216 | 100.0 |

1.46) times higher likelihood of having a normal nutritional condition than being stunted and severely stunted.

Maternal factors that influenced the nutritional status of infant children significantly and statistically were education and working status. Infant and children of non-working mothers were 1.12 (95%CI: 1.04–1.21) more likely to have normal nutritional status than being stunted and severely stunted compared to children of working women. In addition, infant and children of mothers with higher education had higher probability (AOR = 1.50; 95%CI: 1.30–1.72) of being normal than being stunted and severely stunted compared to children of mothers with lower education.

Household characteristics were found to be associated with child's nutritional status statistically and significantly. Infant and children who came from families with the fourth and fifth quintile of household monthly expenditure were, respectively, 1.35 (95%CI: 1.20–1.52) and 1.65 (95%CI: 1.44–1.85) times more likely to have normal nutritional status than to be stunted and severely stunted compared to infant and children who came from families with the first quintile of household monthly expenditure. In addition, the greater the number of

**Table 4. Odds Ratio (OR) of the ordinal logistic regression RISKESDAS 2018.**

| Covariates | | OR [95%CI] | *p*-value |
|---|---|---|---|
| Meet Minimum Dietary Diversity (ref: No) | | | |
| | Yes | 1.15 [1.07–1.24] | < 0.001 |
| Early Initiation of Breastfeeding (ref: No) | | | |
| | Yes | 1.02 [0.95–1.10] | < 0.568 |
| Breastfeeding Status (ref: Currently breastfed) | | | |
| | Ever breastfed | 1.33 [1.22–1.46] | < 0.001 |
| | Never breastfed | 1.01 [0.87–1.17] | 0.939 |
| Child's Age (ref. 18–23 months) | | | |
| | 6–11 months | 3.07 [2.79–3.38] | < 0.001 |
| | 12–17 months | 1.51 [1.39–1.64] | < 0.001 |
| Child's Sex (ref: Male) | | | |
| **Female** | | 1.35 [1.25–1.46] | < 0.001 |
| Mother' (ref. <20 years) | | | |
| 20–29 years | | 1.28 [0.95–1.27] | 0.201 |
| 30+ years | | 1.43 [0.96–1.28] | 0.154 |
| Mother's Working Status (ref. Working) | | | |
| | Not working | 1.12 [1.04–1.21] | < 0.001 |
| Mother's Education (ref. Never Attended School/Primary School) | | | |
| | Junior/Senior High School | 1.25 [1.15–1.36] | < 0.001 |
| | Higher Education | 1.50 [1.30–1.72] | < 0.001 |
| Household Monthly Expenditure Quintile (ref: Lowest) | | | |
| | Second | 1.04 [0.93–0.12] | 0.531 |
| | Middle | 1.11 [0.99–1.24] | 0.071 |
| | Fourth | 1.35 [1.20–1.52] | < 0.001 |
| | Highest | 1.65 [1.44–1.85] | < 0.001 |
| Number of Household Members | | 0.93 [0.91–0.95] | < 0.001 |
| Residence (ref. Rural) | | | |
| | Urban | 1.16 [1.08–1.25] | < 0.001 |
| *Constant 1* | | −0.61 | |
| *Constant 2* | | 0.57 | |

household members, the lower the odds, 0.93 (95%CI: 0.91–0.95) times less likely, of having normal nutritional status than to being stunted and severely stunted. In other words, the smaller the number of household members, the higher the odds, 1.07 (95%CI: 1.05–1.10) times more likely, of having normal nutritional status than to being stunted and severely stunted. Further, children who lived in urban areas were 1.16 (95%CI: 1.08–1.25) times less likely to have normal nutritional status than to be stunted and severely stunted compared to children who lived in rural areas.

## Discussion

The results of this preliminary analysis showed a number of interesting findings. The nutritional status of children was significantly influenced by adherence to the minimum dietary diversity criteria. In comparison to stunted and severely stunted nutritional status, children that met the minimal dietary diversity criteria had a higher chance of in a normal nutritional state. This demonstrates a link between a lack of food diversity and stunting [19]. Diets that include a range of foods encourage children to eat more legumes and nuts, dairy products and their derivatives (dairy products), meats, eggs, vitamin A-rich fruits and vegetables, and other fruits that contribute significantly to their overall nutritional intake, particularly in terms of nutritional micronutrient adequacy [22].

In addition to the availability of food variety, infants and children who are breastfed have a better chance of maintaining a healthy nutritional status. Breast milk delivers a suitable number of calories for infants and contains all of the nutrients required to build and provide the required energy for optimal physical growth [25]. If breast milk is provided from birth, the immune system of the babies who consume the first breast milk contains increased colostrum [25], which helps to avoid infection and disease that can disrupt nutritional absorption and growth [26]. The findings of our study confirmed the importance of feeding practices that meet the criteria of food diversity as well as breastfeeding for the nutritional status of infants and children.

In terms of infants and children demographic characteristics, our study found that as children got older they were less likely to have normal nutritional status. This is due to the fact that as a child grows older, his or her nutritional needs will grow as well. Linear growth issues can occur as a child's nutritional needs increase and he or she does not receive sufficient complementary foods [27]. Moreover, exposure to numerous diseases, such as poor food hygiene and poor environmental cleanliness, is likely to contribute to poor growth as a child grows older [27]. Based on gender, we found that girls were more likely to have a normal nutritional status than to have stunted and severely stunted nutritional status. It is assumed that boys have higher sensitivity to infections and other disorders that can impede their physical development [27].

When it comes to the mother's qualities, mothers with a higher degree of education were also shown to have a higher likelihood of producing healthy babies and children (compared to stunted and severely stunted status). Mothers who have received a good education are more likely to be selective and inventive when it comes to giving nutritional meals for their children, as well as determining the health of their infants [26]. Stunting is likely to be reduced in Indonesia [19] if pregnant women and mothers with children under the age of five get effective instruction on proper eating habits. Thus, it can be concluded that mothers who have higher education have more exposure to information and knowledge about adequate feeding practices and nutritional status.

Working mothers' status was discovered to have an influence, reducing the likelihood of infants and children having a better nutritional status. Working mothers do not have as much time as non-working mothers for nursing, thus they are more likely to stop exclusively

breastfeeding their babies and opt for a more practical solution such as substituting breast milk or giving supplemental formula milk [25, 28].

Households with greater economic standing, particularly those in the fourth (4) and fifth (5) quintiles, and residing in urban areas, had a higher likelihood of having normal nutritional status compared to stunted and severely stunted nutritional status. This results support finding from previous research [15]. Higher socioeconomic level and ease of access in urban areas greatly enables higher-income families to purchase and obtain high-quality food, adequate health services, improved sanitation, and safe drinking water.

Larger household size may diminish the likelihood of babies and children getting a healthy diet. The consumption per capita in the household decreases as the number of household member increases. The risk of stunting depends on the number of children under the age of five in a household. This is linked to the family's ability to meet its members' nutritional demands [19]. Multiple children under the age of five can also lead to inadequate breastfeeding and supplemental feeding practices [19].

This study poses several strengths. First, the dependent variable, which was nutritional status in Indonesia, was only available from this data source. Second, the data used in this study documented comprehensive measures for dietary diversity. Finally, this study was the first nationwide study of the association between dietary diversity and the nutritional status of children aged 6–23 months in Indonesia. The study also has some limitations. First, it was a cross-sectional study. Therefore, it did not permit us to decisively conclude whether the reported stunted condition was caused by the unmet minimum dietary diversity. Second, only dietary diversity was available as the indicator of feeding practices that could be used in the study. Further studies at a sub-national level may enrich our understanding of the relationship between dietary diversity and stunting in different areas. Future researches may also consider to use additional data from other national survey, such as PODES (*Potensi Desa*/Village Potential Survey) which provides more context variables for analysis.

## Conclusion

Stunting is still a main health development problem in Indonesia where almost one third of under five years of age children were stunted or severely stunted. Unmet dietary diversity was among the causes of stunting, together with household and family factors and breastfeeding practices. The findings of this study implied that in order to ensure the achievement of national goal of preventing stunting and sustainable development goal of ending all forms of malnutrition in Indonesia the strategy should promote the fulfillment of minimum food consumption diversity. Information, education, and communication on stunting prevention should be improved so that all mothers not only have knowledge and access to more diverse types of food, but also feed their children according to the minimum dietary diversity.

Efforts to improve the nutritional status of infants and children through feeding must be accompanied by broader activities that go beyond healthcare operations. The government and all relevant stakeholders must be concerned about empowering the household economy and offering alternative access to diverse foods. Improved strategy of stunting prevention should also be targeted to children who are not breastfed, aged 12–23 months, male, children of working mothers, children of lower educated mothers, children from poorer households, children from larger size family, and rural children.

## Author Contributions

**Conceptualization:** Omas Bulan Samosir, Dinda Srikandi Radjiman, Flora Aninditya.

**Data curation:** Omas Bulan Samosir, Dinda Srikandi Radjiman, Flora Aninditya.

**Formal analysis:** Omas Bulan Samosir, Dinda Srikandi Radjiman, Flora Aninditya.

**Funding acquisition:** Omas Bulan Samosir, Dinda Srikandi Radjiman, Flora Aninditya.

**Investigation:** Omas Bulan Samosir, Dinda Srikandi Radjiman, Flora Aninditya.

**Methodology:** Omas Bulan Samosir, Dinda Srikandi Radjiman, Flora Aninditya.

**Project administration:** Omas Bulan Samosir, Dinda Srikandi Radjiman, Flora Aninditya.

**Resources:** Omas Bulan Samosir, Dinda Srikandi Radjiman, Flora Aninditya.

**Software:** Flora Aninditya.

**Supervision:** Omas Bulan Samosir.

**Validation:** Omas Bulan Samosir, Flora Aninditya.

**Visualization:** Omas Bulan Samosir, Flora Aninditya.

**Writing – original draft:** Omas Bulan Samosir, Dinda Srikandi Radjiman, Flora Aninditya.

**Writing – review & editing:** Omas Bulan Samosir, Dinda Srikandi Radjiman, Flora Aninditya.

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
