## [Decision Letter · Decision Letter 0]

2 Sep 2022

PONE-D-21-39580Feeding practices and nutritional status among children aged 6–23 months in Indonesia: the analysis of the results of the 2018 Basic Health ResearchPLOS ONE

Dear Dr. Samosir,

Thank you for submitting your manuscript to PLOS ONE. After careful consideration, we feel that it has merit but does not fully meet PLOS ONE’s publication criteria as it currently stands. Therefore, we invite you to submit a revised version of the manuscript that addresses the points raised during the review process.

We look forward to receiving your revised manuscript.

Kind regards,

Resham B Khatri, PhD

Guest Editor

PLOS ONE

Journal Requirements:

“This research was supported by the Government of Indonesia, which provided for funding for this research project through the Ministry of Research and Technology/National Research and Innovation Board Fiscal Year 2021 (No.: NKB-006/UN2.RST/HKP.05.00/2021) This research was supported by the Government of Indonesia, which provided for funding for this research project through the Ministry of Research and Technology/National Research and Innovation Board Fiscal Year 2021 (No.: NKB-006/UN2.RST/HKP.05.00/2021)”

Reviewers' comments:

Reviewer's Responses to Questions

**Comments to the Author**

1. Is the manuscript technically sound, and do the data support the conclusions?

Reviewer #1: Yes

Reviewer #2: Partly

2. Has the statistical analysis been performed appropriately and rigorously? 

Reviewer #1: Yes

Reviewer #2: Yes

3. Have the authors made all data underlying the findings in their manuscript fully available?

Reviewer #1: Yes

Reviewer #2: Yes

4. Is the manuscript presented in an intelligible fashion and written in standard English?

Reviewer #1: Yes

Reviewer #2: No

5. Review Comments to the Author

Reviewer #1: This manuscript describes findings from a logistic regression study in Indonesia which highlights the association between feeding practices and nutritional status among children in Indonesia. Overall, this is good manuscript that can contribute to the global child health literature. However, some

aspects of the manuscript need to be addressed to strengthen the paper and consider it for

publication, including:

1. Abstract: it is suggested that statistic data (OR, 95%CI, or p-value) should be added to the logistic finding.

2. Introduction and literature review should be compiled and shortened.

3. Please add the average age of the sample

4. Line 343: ‘is’ needs to be replaced by ‘are’

5. Line 358: ‘is to have’ is better changed with ‘has’

6. Line 410-415: statements does not support impact of living in rural and urban area as NTT and NTB cannot define as rural or urban are. Statement needs revision and more supporting literature.

7. Limitation of the study is not written.

8. Line 450: study aim is not necessarily shown up in the conclusion.

9. The conclusion is too long. Suggestion: make it shorter

10. Good recommendation, although lack of suggestion of future research.

Reviewer #2: I am happy to review this paper. The paper titled Feeding practices and nutritional status among children aged 6–23 months in Indonesia: the analysis of the results of the 2018 Basic Health Research is a good attempt to examine the association between feeding practices and nutritional status among child aged 6-23 months in Indonesia. The authors have chosen important research topic of Feeding practices and nutritional status among children aged 6–23 months in Indonesia. However, there are several serious issues with this paper. The introduction is poorly written and research gap is not well documented. The introduction section is too much lengthy and repetitive. Most of the sentences in the introduction part is unclear and redundant. I am concerned about objective; the study objective is not consistent in abstract and introduction section. Also, though the title of the study is Feeding practices and nutritional status among children aged 6–23 months but the authors have only taken food diversity as part of the feeding practices so I think the title of the study may require some revision; food diversity and nutritious status rather than feeding practices and nutrition status.

Massive work is required in the introduction section to make it succinct and ensure coherence. The author has added literature review part in separate heading under introduction, but I think it is not needed separately rather some important points can be combined in the intro part and rest can be omitted. Also, a clear justification is required in the introduction section, to justify how this paper adds value. In result section, the interpretation of the finding is too much lengthy and inappropriate, I strongly suggest author to take reference of other PLoS one paper to write the interpretation in an appropriate way. The discussion section needs to be revised ensure proper and consistent flow. The authors have not discussed about the strengths and limitation of their study. Also, the manuscript needs massive English edits and reduce errors.

This manuscript requires serious formatting work to make as per journal requirement. I strongly suggest authors to seriously review the journal guideline and revise it accordingly. The conclusion and recommendation should be based on the findings.

Abstract:

The author has explained the result part only in the narrative form and it doesn’t contain any data so adding some key value of Odds would improve the paper. Similarly, in the abstract, the conclusion part is very vague and so suggested to revise it based on your findings.

Introduction part:

In the abstract author said the objective of the study was to investigate the association between feeding practices and nutrition status of children and in the introduction section; author has mentioned the objective of this study was to investigate the association between the food consumption diversity and nutritional status of children.

I am concerned about the objective of the study, whether authors assess overall feeding practices and nutrition status or only food diversity and nutrition status. Please make it consistent. Also, if you are primarily interested to assess food diversity and nutrition status then you are suggested to revise your research title accordingly.

The introduction section is too much lengthy and repetitive. Most of the sentences in the introduction part is unclear and redundant. I noticed the author has mentioned one figure in the introduction part I suggested to remove figure from the intro part.

I am concerned why author has added so much long literature review part under separate heading in the introduction section, rather some important points can be combined in the intro part and rest can be omitted to make intro part succinct. The massive work is required in the introduction part and The English editing is highly required.

Methods

The author has not discussed about study design and sampling methods, they have not clearly described how they categorized sociodemographic variables in the method section. There are several grammatic errors in the sentence construction in the methodology part. Authors are suggested to add, how did they ensure quality control during overall study.

Result

The interpretation of the tables is not appropriate, please refer the similar PLoS publications to get an idea to describe the findings in the paper. Along with the narrative, the value of the key findings should be mentioned so that reader can relate value with the narrative, and it makes sense to the readers. Also, the interpretation of the table is too much lengthy. In fact, you do not need to interpret whole things from the table rather you can explain some key observations and rest reader can refer to the table. So, I strongly suggest author to revise your interpretation of the tables in the result section and make it more succinct and appropriate.

Discussion and conclusion:

The discussion section needs to be revised ensure proper and consistent flow. The authors have not discussed about strengths and limitation of the study. The conclusion section is very vague and not specific to the overall study findings. The discussion and conclusion section need to be revised based on how the authors would address my comments.

6. PLOS authors have the option to publish the peer review history of their article (what does this mean?). If published, this will include your full peer review and any attached files.

Reviewer #1: **Yes: **Dyah Juliastuti

Reviewer #2: No

---

## [Author Response · Author response to Decision Letter 0]

24 Oct 2022

RESPONSE TO REVIEWERS

Reviewer #1

No. Comment Response

1. This manuscript describes findings from a logistic regression study in Indonesia which highlights the association between feeding practices and nutritional status among children in Indonesia. Overall, this is good manuscript that can contribute to the global child health literature. However, some

aspects of the manuscript need to be addressed to strengthen the paper and consider it for

publication, including: Thank you very much for the constructive comment.

Some aspects of the manuscript have been addressed to strengthen the paper and consider it for publication.

2. Abstract: it is suggested that statistic data (OR, 95%CI, or p-value) should be added to the logistic finding. Thank you very much for the constructive suggestion.

The adjusted odds ratio (AOR) and 95% CI have been added in the Abstract.

3. Introduction and literature review should be compiled and shortened. Thank you very much for the constructive comment.

Introduction and literature review has been compiled and shortened.

4. Please add the average age of the sample. Thank you very much for the constructive suggestion.

The average age of the sample has been added in line 224-225.

5. Line 343: ‘is’ needs to be replaced by ‘are’ Thank you very much for the constructive suggestion.

The whole sentence has been dropped during the revision process.

6. Line 358: ‘is to have’ is better changed with ‘has’ Thank you very much for the constructive suggestion.

Line 358 now becomes line 307 and has been rewritten.

7. Line 410-415: statement does not support impact of living in rural and urban area as NTT and NTB cannot define as rural or urban are. Statement needs revision and more supporting literature. Thank you very much for the constructive comment.

The statement has been deleted.

8. Limitation of the study is not written. Thank you very much for the constructive comment.

Limitation of the study has been written in line 347-350.

9. Line 450: study aim is not necessarily shown up in the conclusion. Thank you very much for the constructive suggestion.

Study aim has been deleted from the conclusion. 

10. The conclusion is too long. Suggestion: make it shorter. Thank you very much for the constructive suggestion.

The conclusion has been made shorter.

11. Good recommendation, although lack of suggestion of future research. Thank you very much for the constructive comment.

Suggestion for further research has been added in line 350-354.

Reviewer #2

No. Comment Response

1. I am happy to review this paper. The paper titled Feeding practices and nutritional status among children aged 6–23 months in Indonesia: the analysis of the results of the 2018 Basic Health Research is a good attempt to examine the association between feeding practices and nutritional status among child aged 6-23 months in Indonesia. The authors have chosen important research topic of Feeding practices and nutritional status among children aged 6–23 months in Indonesia. However, there are several serious issues with this paper. Thank you very much for the constructive comment.

2. The introduction is poorly written and research gap is not well documented. Thank you very much for the constructive comment.

The introduction has been revised and research gap has been documented in line 136-137.

3. The introduction section is too much lengthy and repetitive. Most of the sentences in the introduction part is unclear and redundant. Thank you very much for the constructive comment.

The introduction section has been revised, shortened, and made clear.

4. I am concerned about objective; the study objective is not consistent in abstract and introduction section. Also, though the title of the study is Feeding practices and nutritional status among children aged 6–23 months but the authors have only taken food diversity as part of the feeding practices so I think the title of the study may require some revision; food diversity and nutritious status rather than feeding practices and nutrition status. Thank you very much for the constructive comment and suggestion.

The title of the study has been changed to “Food consumption diversity and nutritional status among children aged 6–23 months in Indonesia: the analysis of the results of the 2018 Basic Health Research” and the objective of the study has been revised accordingly.

5. Massive work is required in the introduction section to make it succinct and ensure coherence. The author has added literature review part in separate heading under introduction, but I think it is not needed separately rather some important points can be combined in the intro part and rest can be omitted. Thank you very much for the constructive comment and suggestion.

The introduction and literature review have been combined.

6. Also, a clear justification is required in the introduction section, to justify how this paper adds value. Thank you very much for the constructive suggestion.

The clear justification and novelty of the study have been added in line 136-137.

7. In result section, the interpretation of the finding is too much lengthy and inappropriate, I strongly suggest author to take reference of other PLOS ONE paper to write the interpretation in an appropriate way. Thank you very much for the constructive comment and suggestion.

The interpretation has been revised by taking reference of other PLOS ONE paper to write the interpretation in an appropriate way. 

8. The discussion section needs to be revised ensure proper and consistent flow. Thank you very much for the constructive suggestion.

The discussion section has been revised to ensure proper and consistent flow.

9. The authors have not discussed about the strengths and limitation of their study. Thank you very much for the constructive comment.

The strengths and limitation of the study have been added in line 343-350.

10. Also, the manuscript needs massive English edits and reduce errors. Thank you very much for the constructive comment.

The manuscript has been improved.

11. This manuscript requires serious formatting work to make as per journal requirement. I strongly suggest authors to seriously review the journal guideline and revise it accordingly. Thank you very much for the constructive comment.

The manuscripts has been revised and improved.

12. The conclusion and recommendation should be based on the findings.

 Thank you very much for the constructive suggestion.

The conclusion and recommendation have been revised based on the findings.

 

No. Comment Response

1. Abstract:

The author has explained the result part only in the narrative form and it doesn’t contain any data so adding some key value of Odds would improve the paper. 

Similarly, in the abstract, the conclusion part is very vague and so suggested to revise it based on your findings. Thank you very much for the constructive comment.

The adjusted odds ratio (AOR) and 95% CI have been added in the Abstract.

The conclusion has been revised.

2. Introduction part:

In the abstract author said the objective of the study was to investigate the association between feeding practices and nutrition status of children and in the introduction section; author has mentioned the objective of this study was to investigate the association between the food consumption diversity and nutritional status of children. Thank you very much for the constructive suggestion.

The adjusted odds ratio (AOR) and 95% CI have been added in the Abstract.

3. I am concerned about the objective of the study, whether authors assess overall feeding practices and nutrition status or only food diversity and nutrition status. Please make it consistent. Also, if you are primarily interested to assess food diversity and nutrition status then you are suggested to revise your research title accordingly. Thank you very much for the constructive comment and suggestion.

The title of the study has been changed to “Food consumption diversity and nutritional status among children aged 6–23 months in Indonesia: the analysis of the results of the 2018 Basic Health Research” and the objective of the study has been revised accordingly.

4. The introduction section is too much lengthy and repetitive. Most of the sentences in the introduction part is unclear and redundant. I noticed the author has mentioned one figure in the introduction part I suggested to remove figure from the intro part. Thank you very much for the constructive comment and suggestion.

The introduction section has been revised.

Figure 1 has been removed from the introduction part.

5. I am concerned why author has added so much long literature review part under separate heading in the introduction section, rather some important points can be combined in the intro part and rest can be omitted to make intro part succinct. The massive work is required in the introduction part and The English editing is highly required. Thank you very much for the constructive comment and suggestion.

The introduction and literature section have been revised and combined.

6. Methods

The author has not discussed about study design and sampling methods. They have not clearly described how they categorized sociodemographic variables in the method section. There are several grammatical errors in the sentence construction in the methodology part. Authors are suggested to add, how did they ensure quality control during overall study. Thank you very much for the constructive comment and suggestion.

Study design and sampling methods of the survey have been added.

Grammatical errors in the sentence construction in the methodology part have been corrected.

7. Result

The interpretation of the tables is not appropriate, please refer the similar PLOS ONE publications to get an idea to describe the findings in the paper. Along with the narrative, the value of the key findings should be mentioned so that reader can relate value with the narrative, and it makes sense to the readers. Also, the interpretation of the table is too much lengthy. In fact, you do not need to interpret whole things from the table rather you can explain some key observations and rest reader can refer to the table. So, I strongly suggest author to revise your interpretation of the tables in the result section and make it more succinct and appropriate. Thank you very much for the constructive comment and suggestion.

The interpretation of the tables have been revised.

8. Discussion and conclusion:

The discussion section needs to be revised ensure proper and consistent flow. The authors have not discussed about strengths and limitation of the study. The conclusion section is very vague and not specific to the overall study findings. The discussion and conclusion section need to be revised based on how the authors would address my comments. Thank you very much for the constructive comment and suggestion.

The discussion section has been revised.

The strengths and limitation of the study have been added in line 343-350.

The conclusion section has been revised based on the study findings. 

The discussion and conclusion section have been revised based on the findings.

---

## [Decision Letter · Decision Letter 1]

7 Dec 2022

PONE-D-21-39580R1Food consumption diversity and nutritional status among children aged 6–23 months in Indonesia: the analysis of the results of the 2018 Basic Health ResearchPLOS ONE

Dear Dr. Samosir,

Thank you for submitting your manuscript to PLOS ONE. After careful consideration, we feel that it has merit but does not fully meet PLOS ONE’s publication criteria as it currently stands. Therefore, we invite you to submit a revised version of the manuscript that addresses the points raised during the review process.

We look forward to receiving your revised manuscript.

Kind regards,

Resham B Khatri, PhD

Guest Editor

PLOS ONE

Journal Requirements:

Reviewers' comments:

Reviewer's Responses to Questions

**Comments to the Author**

1. If the authors have adequately addressed your comments raised in a previous round of review and you feel that this manuscript is now acceptable for publication, you may indicate that here to bypass the “Comments to the Author” section, enter your conflict of interest statement in the “Confidential to Editor” section, and submit your "Accept" recommendation.

Reviewer #1: All comments have been addressed

Reviewer #2: All comments have been addressed

2. Is the manuscript technically sound, and do the data support the conclusions?

Reviewer #1: Yes

Reviewer #2: Yes

3. Has the statistical analysis been performed appropriately and rigorously? 

Reviewer #1: Yes

Reviewer #2: Yes

4. Have the authors made all data underlying the findings in their manuscript fully available?

Reviewer #1: Yes

Reviewer #2: Yes

5. Is the manuscript presented in an intelligible fashion and written in standard English?

Reviewer #1: Yes

Reviewer #2: No

6. Review Comments to the Author

Reviewer #1: Thank you for the opportunity to re-review this interesting article. According to my previous review, the author has already revised the article per suggested, which I appreciate. My only concern is that in the ethics statement, the author indicated as if he is the one who conduct the data collection which is not congruent with the statement in line 146 which shows that the author used secondary data from the basic health research. It is suggested that the reference of the ethics process is added, and permit to use the Riskesdas data is mentioned.

Reviewer #2: Dear authors, thank you so much for addressing all of my comments. The manuscript has improved now.

The introduction is concise and methodology section is well-explained.

The results are arranged systematically and coherently narrated and conclusion aligns with the findings.

However, I have a few suggestions that may further improve manuscript.

I think thorough editing of the english langauge would enhance the manuscript and minimize grammatical and sentence erros.

I noticed that authors used the present tense where it should have been be past; for instances; line no. 35, 66-67, 87, 100-please change these sentence to the past tense.

In line no. 64-65 reference is missing “ In the Southeast Asia region, the prevalence of stunting was the second highest 65 in Indonesia after Timor-Leste”[....Ref..]

Please check the references again to make sure they adhere to journal standards.

The manuscript requires some formatting work.

Thank you

7. PLOS authors have the option to publish the peer review history of their article (what does this mean?). If published, this will include your full peer review and any attached files.

Reviewer #1: No

Reviewer #2: **Yes: **Dipendra Singh Thakuri

---

## [Author Response · Author response to Decision Letter 1]

12 Jan 2023

Dear Dr. Resham B Khatri,

Thank you for your and the reviewers’ reply regarding our manuscript “Food consumption diversity and nutritional status among children aged 6–23 months in Indonesia: the analysis of the results of the 2018 Basic Health Research”. We are grateful for the reviewer’s comments and the positive valuation of our work. We have revised and modified the manuscript, as shown in the table below. 

Reviewer #1

Feedback Response

“My only concern is that in the ethics statement, the author indicated as if he is the one who conduct the data collection which is not congruent with the statement in line 146 which shows that the author used secondary data from the basic health research. It is suggested that the reference of the ethics process is added, and permit to use the Riskesdas data is mentioned.”

 Thank you for flagging this. 

Yes, RISKESDAS survey was implemented by the Ministry of Health. We have edited the ethics statement accordingly. 

Reviewer #2

Feedback Responses

“I think thorough editing of the English language would enhance the manuscript and minimize grammatical and sentence errors.”

 Thank you for your comments.

We have revised present tenses into past tenses.

“I noticed that authors used the present tense where it should have been be past; for instances; line no. 35, 66-67, 87, 100-please change these sentence to the past tense.” Thank you for your input. We have edited these parts into past tenses.

“In line no. 64-65 reference is missing “ In the Southeast Asia region, the prevalence of stunting was the second highest 65 in Indonesia after Timor-Leste”[....Ref..]”

 Thank you for pointing this out. We have added the reference number for this part. 

“Please check the references again to make sure they adhere to journal standards. The manuscript requires some formatting work.”

 Thank you for your suggestions. We have edited several parts of the reference list as PLOS ONE’s standards.

All the changes above have improved the manuscript, and we hope it can meet PLOS ONE’s publication criteria. 

Sincerely, 

Authors: Omas Bulan Samosir, Dinda Srikandi Radjiman, and Flora Aninditya

---

## [Editor Report · Decision Letter 2]

24 Jan 2023

Food consumption diversity and nutritional status among children aged 6–23 months in Indonesia: the analysis of the results of the 2018 Basic Health Research

PONE-D-21-39580R2

Dear Dr. Samosir,

We’re pleased to inform you that your manuscript has been judged scientifically suitable for publication and will be formally accepted for publication once it meets all outstanding technical requirements.

Kind regards,

Resham B Khatri, PhD

Guest Editor

PLOS ONE
---

## [Editor Report · Acceptance letter]

26 Jan 2023

PONE-D-21-39580R2 

Food consumption diversity and nutritional status among children aged 6–23 months in Indonesia: the analysis of the results of the 2018 Basic Health Research 

Dear Dr. Samosir:

I'm pleased to inform you that your manuscript has been deemed suitable for publication in PLOS ONE. Congratulations! Your manuscript is now with our production department. 

Kind regards, 

on behalf of

Dr. Resham B Khatri 

Guest Editor

PLOS ONE